# THE BEST OF BOTH WORLDS: IMPROVED OUTCOME PREDICTION USING CAUSAL STRUCTURE LEARNING

## ABSTRACT

In limited data settings as in the medical domain, causal structure learning can be a powerful tool for understanding the relationships between variables and achieving out of sample generalisation for the prediction of a specific target variable. Most methods that learn causal structure from observational data rely on strong assumptions, such as the absence of unmeasured confounders, that are not valid in real world scenarios. In addition, due to evolving conditions and treatment approaches, causal relationships between the variables change over time. Moreover in a clinical setting, symptoms often need to be managed before finding the root cause of a problem, which puts the emphasis on accurate outcome prediction. Consequently, prediction of a specific target variable from retrospective observational data based on causal relationships alone will not be sufficient for generalisation to prospective data. To overcome these limitations, we opt for ***the best of both worlds*** in this work by learning a shared representation between causal structure learning and outcome prediction. We provide extensive empirical evidence to show that this would not only facilitate out-of-sample generalisation in outcome prediction but also enhance robust causal discovery for the outcome variable. We also highlight the strengths of our model in terms of time efficiency and interpretability. Code is available at:

## 1 INTRODUCTION

Personalised medicine is a branch of medicine, which aims at providing individualised therapy based on patient's phenotype. This is a closed loop process involving analyses of treatment response or outcomes, and treatment adjustment. The treatment response and patient outcomes are influenced by various factors such as disease characteristics, patient traits and the environment (Ambrosone et al., 2006). Randomised controlled trials (RCTs) enable the prospective evaluation of treatment response in randomised groups of patients under controlled conditions. Therefore, RCTs provide a reliable means to assess the cause-effect relationships between treatments and outcomes by eliminating confounding bias (Hariton & Locascio, 2018; Bhide et al., 2018). However, it is not always feasible to conduct RCTs as they can be time-consuming, expensive and suffer from post-randomisation biases (Fernainy et al., 2024).

In recent times, machine learning-based methods have been successfully used for the prediction of patient outcomes based on observational data (Lee et al., 2021; 2024; Babaei Rikan et al., 2024; Alaa et al., 2017). Most modern machine learning methods find linear or non-linear associations between observational data and outcome. As the associations are learnt on a sample of the data, larger sample sizes increase the generalisability of the associations to unseen samples of the data (Chekroud et al., 2024). Ideally, these methods are evaluated using observed outcomes or expert annotations, which are both time expensive. Consequently, data in the medical domain is limited, unstructured and incomplete. This in turn makes generalisation to out-of-sample data more difficult for machine learning based outcome prediction methods (Goetz et al., 2024). The lack of transparency in some of these machine learning methods makes them difficult to interpret, compounding the challenges.

Causal structure learning is concerned with learning causal relationships from observational data. Popular techniques employ machine learning methods to model the causal relationships between the variables of observational data by imposing certain topological constraints (Zheng et al., 2018; Yu et al., 2019; Ng et al., 2019). Causal structure learning methods have the potential to improve

interpretability in the medical domain by finding causal relationships between observed variables and the outcome for various downstream analyses (Feuerriegel et al., 2024; Piccininni et al., 2020). Consequently, they can bridge the gap between observational studies and RCTs. However, most of these methods make strong assumptions about the data, which might not be valid in a real world setting (Montagna et al., 2024).

One such assumption is the absence of unmeasured confounders (Kalisch & Bühlman, 2007; Shimizu et al., 2006). This is not realisable without domain knowledge or time-expensive expert intervention (Bica et al., 2021). Moreover, patient outcomes are also influenced by evolving knowledge, treatment approaches and the environment (Futoma et al., 2020; Petzschner, 2024). Consequently, relying solely on causal relationships to predict outcomes presents a significant challenge for generalizing methods to prospective data, as these associations are derived from retrospectively observed data. This limitation underscores the difficulty of ensuring that findings translate effectively to future scenarios.

We overcome these limitations in this work by opting for *the best of both worlds* — causal structure learning and machine learning-based outcome prediction. Our contributions are as follows.

• We designed our approach to learn outcome prediction and causal structure simultaneously. In our framework, causal structure learning functions as an auxiliary task to support outcome prediction, sharing representations of the input through the hidden layers of our network architecture and employing task-specific heads for refined predictions.

• We provide empirical evidence to show that this learning strategy enables (i) interpretability by visualisation of the learnt causal graph (ii) out-of-sample generalisation for outcome prediction. The primary focus of our work is to improve generalisation for outcome prediction in the medical domain. Causal structure learning functions as an auxiliary task to support outcome prediction. Despite this, we also provide evidence demonstrating the benefits of our approach in robust causal discovery for the outcome variable.

• We provide a case study by applying the method to survival analysis. We show that the proposed framework improves interpretability of the model and generalisability to unseen data in real world scenarios. We also comment on the clinical relevance of the results.

## 2    RELATED WORK

Most causal structure learning methods have been developed to learn causal relationships from observational data based on the foundations of causal graphical model (Pearl, 2009). These methods can be classified into three broad categories: (i) constraint based methods that use conditional independence tests to infer the direction of causal relationship between variables (Kalisch & Bühlman, 2007), (ii) methods that use functional causal models to identify the causal structure by making assumptions about the data distribution (Shimizu et al., 2006), (iii) score-based methods which either adopt greedy search algorithms to determine the causal structure (Chickering, 2002) or impose topological constraints to learn the causal structure (Zheng et al., 2018). Most of these methods make strong assumptions about the data (Montagna et al., 2024) which might not be realisable in a real world setting.

Recent works (Kyono et al., 2020; Ge et al., 2023) use causal structure learning to improve generalisation in supervised learning. Ge et al. (2023) build upon (Zheng et al., 2018) to learn robust causal structures that are invariant to the data environments by getting rid of spurious correlations arising from the data environment. This is contradictory to our aim of banking on the rich information from evolving conditions to predict the target. Kyono et al. (2020) introduce a causal structure learning based regularizer, CASTLE, for improving generalisation in supervised learning. They add a supervised loss term to the non-linear framework from Zheng et al. (2018) to learn the target variable. CASTLE (Kyono et al., 2020) is one of the revolutionary works which demonstrated the superior performance of causality based regularaisation over commonly used regularisation techniques for deep learning such as L1-norm, L2-norm, dropout and early stopping (Tibshirani, 1996; Hoerl & Kennard, 1970; Goodfellow et al., 2016).

However, we observe several unsolved challenges of this work: (i) the feed-forward architecture used by CASTLE does not scale with the feature variables, (ii) CASTLE treats the target variable

reconstructed as a part of causal structure learning as the final output which hinders not only causal structure learning but also outcome prediction. We address these research gaps by (i): adopting a graph autoencoder-based causal structure learning method (Ng et al., 2019), which builds a single graph for all the variables and scales with the number of features, (ii) we introduce an additional task-specific head for outcome prediction, which exploits the representation shared with causal structure learning to reliably predict the outcome and generalise well to unseen data.

## 3 CAUSAL STRUCTURE LEARNING

Given observational data $\mathbf{X} \in \mathbb{R}^{n \times d}$, consisting of $n$ i.i.d. samples of the random vector $X = (X_1, X_2, ....X_d)$, score based methods learn an optimal causal directed acyclic graph (DAG), $\mathcal{G}(\mathbf{W})$ on $d$ nodes from a discrete space of DAGs $\mathbb{D}$ for the joint distribution $\mathbb{P}(X)$ (Spirtes et al., 2001). Here, X is modelled by considering the data generating process in a linear structural equation model (SEM) defined by the weighted adjacency matrix $\mathbf{W} \in \mathbb{R}^{d \times d}$ as in (Hoyer et al., 2008),

$$X_j := W_j^T \mathbf{X} + Z_j$$

for $j = 1, 2, ..., d$; $Z = (Z_1, Z_2, ..., Z_d)$ is a random noise vector. Zheng et al. (2018) impose smooth acylicity constraint on $\mathbf{W}$ and convert the combinatorial optimisation problem of finding $\mathcal{G}(\mathbf{W}) \in \mathbb{D}$ to a continuous one:

$$\min_{\mathbf{W}} \frac{1}{2n} \sum_{i=1}^{n} \left\| X^{(i)} - \mathbf{W}^T X^{(i)} \right\|_F^2 + \lambda \left\| \mathbf{W} \right\|_1 \tag{1}$$

$$\text{subject to } \text{tr}(e^{\mathbf{W} \odot \mathbf{W}}) - d = 0,$$

where $e^{\mathbf{M}}$ denotes the matrix exponential of $\mathbf{M}$, $\odot$ denotes the Hadamard product and n is the sample size. The L1 regularization term $\left\| \mathbf{W} \right\|_1$ encourages sparsity in the learnt DAGs.

## 4 THE BEST OF BOTH WORLDS - PARADIGM

(Ng et al., 2019) generalises the formulation in (1) to the non-linear case and draws parallels to the graph autoencoder (GAE) framework (Cen et al., 2019). For the linear case, we can rewrite $\mathbf{W}^T X^{(i)}$ in (1) as $\mathbf{W}^T X^{(i)} = f(X^{(i)}, \mathbf{W})$, where $f$ is the data generating model with parameters $\Theta$. Ng et al. (2019) extends this to the non-linear case by considering:

$$f(X^{(i)}, \mathbf{W}) = g_2(\mathbf{W}^T g_1(X^{(i)})), \tag{2}$$

where each variable $X^{(i)}$ is vector valued, i.e., $X^{(i)} \in \mathbb{R}^l$; $g_1 : \mathbb{R}^l \rightarrow \mathbb{R}^l$ and $g_2 : \mathbb{R}^l \rightarrow \mathbb{R}^l$ are Multilayer Perceptrons (MLPs) with shared weights across all variables $X_j$. The formulation in (2) is considered similar to the GAE framework, if we view $g_1$ and $g_2$ as variable-wise encoder and decoder modules and $\mathbf{W}^T g_1(X^{(i)})$ as a linear transformation of the latent representation. The dimension of the latent representation can be adjusted based on the intrinsic dimension of $\mathbf{X}$. We refer to this framework of Ng et al. (2019) as CausalGAE framework. Let $\hat{X}^{(i)} = g_2(\mathbf{W}^T g_1(X^{(i)}))$ be the reconstructed output and $\Theta_1$, $\Theta_2$ be the parameters of $g_1$ and $g_2$ respectively. Then, the framework optimises the following objective function:

$$\min_{\mathbf{W}, \Theta_1, \Theta_2} \frac{1}{2n} \sum_{i=1}^{n} \left\| X^{(i)} - \hat{X}^{(i)} \right\|_F^2 + \lambda \left\| \mathbf{W} \right\|_1 \tag{3}$$

$$\text{subject to } \text{tr}(e^{\mathbf{W} \odot \mathbf{W}}) - d = 0,$$

We build upon this work to derive a formulation for simultaneous causal structure learning and outcome prediction. We consider outcome prediction as a supervised learning task that is concerned

with predicting $Y$ from $\tilde{\mathbf{X}} := (X_1, X_2, ..., X_{d-1}) \in \mathbb{R}^{n \times d}$ variables. We consider $\mathbf{X}_{(d-1)}$ to be the outcome variable $Y$. The formulation in (2) would then restrict approximation of $Y$ to a non-linear function of its causal parents. Outcome prediction is a complex task that is dependent on dynamically changing variables and environment. This will lead to relationships in the data that are not explained by the causal structure alone but are necessary to predict the outcome. Therefore, we hypothesize that a non-linear function of the target variable's causal parents alone is not sufficient to approximate $Y$ and propose the following:

$$\hat{Y}^{(i)} = g_3(\mathbf{W}^T g_1(X^{(i)})), \tag{4}$$

where $g_3$ is a variable-wise non-linear function with parameters $\Theta_3$. In the case of classification, $g_3$ is a projection layer $g_3 : \mathbb{R}^d \to \mathbb{R}^c$, where c is the number of classes. For simplicity, we consider variable $X^{(i)} \in \mathbb{R}^l$ to be scalar valued, i.e., $l = 1$. To summarise, we use $g_1$, a variable-wise encoder to learn a latent representation of the data, $H$. Next, we perform a linear transformation of $H$ using the weighted adjacency matrix $\mathbf{W}$ to produce $\hat{H}$. We then feed $\hat{H}$ to task specific variable-wise decoders $g_2$ and $g_3$ to provide reconstructed output $\hat{X}$ and $\hat{Y}$ respectively. The same is explained in the following:

$$H^{(i)} = g_1(X^{(i)})$$
$$\hat{H}^{(i)} = \mathbf{W}^T g_1(X^{(i)})$$
$$\hat{X}^{(i)} = g_2(\mathbf{W}^T g_1(X^{(i)}))$$
$$\hat{Y}^{(i)} = g_3(\mathbf{W}^T g_1(X^{(i)}))$$

We learn the parameters of the shared encoder and target specific decoders jointly by optimising the following objective function:

$$\min_{\mathbf{W}, \Theta_1, \Theta_2, \Theta_3} \frac{(1-\kappa)}{2n} \sum_{i=1}^n \left\| X^{(i)} - \hat{X}^{(i)} \right\|_F^2 + \lambda \left\| \mathbf{W} \right\|_1 + \frac{\kappa}{n} \sum_{i=1}^n \left\| Y^{(i)} - \hat{Y}^{(i)} \right\|_F^2 \tag{5}$$

$$\text{subject to } \text{tr}(e^{\mathbf{W} \odot \mathbf{W}}) - d = 0,$$

where $\kappa$ is a hyperparameter that can be tuned depending on the dataset and the outcome prediction task. A sensitivity analysis for the same can be found in Appendix C. We use Augmented Lagrangian method to optimise the constrained optimisation problem in (5) (Appendix A). The simplified form of the loss is as follows:

$$\mathcal{L}_\rho(\mathbf{W}, \Theta_1, \Theta_2, \Theta_3, \alpha) = (1-\kappa)\mathcal{L}_{DAG}(\mathbf{W}, \Theta_1, \Theta_2) + \kappa\mathcal{L}_{sup}(\mathbf{W}, \Theta_1, \Theta_3), \tag{6}$$

where $\alpha$ is the Lagrange multiplier. For classification, we use cross entropy loss as the supervised loss.

## 5 RESULTS

We present empirical evidence of our model's generalization performance through a series of experiments on both synthetic and real-world datasets, as detailed below. Furthermore, we demonstrate the model's ability to learn robust representations while enhancing interpretability and scalability. Through a case study, we also show the clinical relevance of our model.

**Experimental setup.** We perform the experiments by splitting the data into 90% training and 10% test datasets. The training dataset is used in a 10-fold cross validation setting to train the models. We choose CASTLE network (Kyono et al., 2020), which has outperformed various regularisation methods like like dropout, data augmentation and batch normalisation, as our primary baseline. We also compare our method with Multilayer Perceptron (MLP) (Pedregosa et al., 2011) and its

regularised variants by employing L2-norm with early stopping (L2+ES) based on training loss, and L2-norm with early stopping based on validation score (ES).

All the methods used Adam optimiser with a learning rate of 1e-3. The models were trained for 300 epochs with an early stopping criterion based on validation loss (except the L2-norm with early stopping (ES) variant of the MLP which employed early stopping based on validation score). In addition, our model and CausalGAE update Lagrange multiplier $\alpha$ and penalty $\rho$ over 20 iterations with early stopping based on a threshold. For both models, we use the default threshold and loss hyperparameters as in (Ng et al., 2019). The loss hyperparameter $\kappa$ is set to 0.25 for our model upon doing a sensitivity analysis (Appendix C). All methods used the same data splits and identical seeds. For all our experiments we use a machine equipped with Intel i9-10900X processor and NVIDIA RTX2080 GPUs. Our code will be publicly available at:

## 5.1 GENERALISATION PERFORMANCE ON SYNTHETIC DATA

We study the generalisation performance of the models on synthetic data for the regression task. We generate synthetic data as in (Ng et al., 2019). We generate a random DAG having $n = 1000$ samples, $d = 20$ nodes and degree of freedom $dof = 3$ from Erdős–Rényi graph. The variables or features $\mathbf{X}$ are sampled from the Additive noise model (ANM) under two conditions described as follows.

**Case 1.** Non-linear causal relationships between the variables as:

$$\mathbf{X} = 2\sin(\mathbf{W}^T(\mathbf{X} + \mathbf{1}) + 0.5 \cdot \mathbf{1}) + \cos(\mathbf{W}^T(\mathbf{X} + \mathbf{1}) + 0.5 \cdot \mathbf{1}) + \mathbf{Z}$$

**Case 2.** We consider $\mathbf{X}_{(d-1)}$ to be the outcome variable $Y$. In addition to the causal relationships described in Case 1, we simulate the case where the outcome is not dependent only on its causal parents by adding an extra term to $Y$ as:

$$Y = 2\sin(\mathbf{W}^T(\mathbf{X} + \mathbf{1}) + 0.5 \cdot \mathbf{1}) + \cos(\mathbf{W}^T(\mathbf{X} + \mathbf{1}) + 0.5 \cdot \mathbf{1}) + \mathbf{Z}$$
$$+ \cos((X^{non-pa(Y)_0} + \mathbf{1}) + 0.5 \cdot \mathbf{1}),$$

where $\mathbf{W}$ is the weighted Adjacency matrix of the random DAG, $\mathbf{Z}$ is additive noise and $X^{non-pa(Y)_0}$ is the first non-parent of $Y$.

Mean squared error (MSE) is chosen as the metric to compare the performance of the models. Table 1 shows the results on test dataset. For this experiment, we also compare our model with CausalGAE (Ng et al., 2019), the causal structure learning framework upon which our model is built. We infer the reconstructed target from the trained model and report the MSE for the test data in Table 1. Our model performs the best in both the cases illustrating its ability to learn not only from causal parents but also from other predictors of the target variable.

## 5.2 ROBUST CAUSAL DISCOVERY

The primary focus of our work is to improve generalisation for outcome prediction in the medical domain. Causal structure learning functions as an auxiliary task to support outcome prediction. Here we demonstrate the performance of our model in recovering true causal graph. We use the synthetic datasets from Case 1 and Case 2, and compare the performance of our model with CausalGAE. We focus our comparison on CausalGAE for two main reasons: (i) our model is built on this framework, and (ii) CausalGAE has demonstrated superior performance compared to widely-used linear and graph neural network (GNN)-based causal structure learning methods, such as NOTEARS (Zheng et al., 2018) and DAG-GNN (Yu et al., 2019), in synthetic data experiments, particularly in Case 1.

We report the false discovery rate (FDR), true positive rate (TPR), false positive rate (FPR) and structural Hamming distance (SHD). As seen in Table 2, our model recovers the true causal graph with SHD comparable to that of CausalGAE. In both cases, our model improves upon TPR. In contrast to our model, CausalGAE fails to identify the causal parents of $Y$ in both the cases as seen in the associated causal graphs included in Appendix D. These results highlight the versatility of the shared representations learned by our model.

Table 1: Generalisation performance on synthetic datasets. Baseline: MLP; L2+ES: MLP + L2-norm with early stopping based on training loss; ES: MLP + L2 norm + early stopping based on validation MSE; Test MSE along with the gap between train and test MSE ($\Delta :=$ Mean $\text{MSE}_{test}$ - Mean $\text{MSE}_{train}$) are reported.

|  | **Case 1** | **Case 2** |
|---|---|---|
| Baseline | $1.141 \pm 0.042$ (0.854) | $0.991 \pm 0.043$ (0.704) |
| L2+ES | $1.091 \pm 0.029$ (0.734) | $0.882 \pm 0.037$ (0.350) |
| ES | $0.974 \pm 0.018$ (0.233) | $0.857 \pm 0.022$ (0.145) |
| CASTLE | $1.073 \pm 0.108$ (0.613) | $0.923 \pm 0.089$ (0.466) |
| CausalGAE | $1.172 \pm 0.000$ (0.164) | $1.027 \pm 0.000$ (0.148) |
| Ours | $\mathbf{0.938 \pm 0.029}$ **(0.086)** | $\mathbf{0.815 \pm 0.029}$ **(0.002)** |

Table 2: Performance of the models in recovering true causal graphs. FDR: false discovery rate; TPR: true positive rate; FPR: false positive rate and SHD: structural Hamming distance.

|  | **FDR** | **TPR** | **FPR** | **SHD** |
|---|---|---|---|---|
| Case 1: CausalGAE | 0.59 | 0.27 | 0.06 | 26 |
| Case 2: CausalGAE | 0.40 | 0.23 | 0.02 | 23 |
| Case 1: Ours | 0.59 | 0.62 | 0.14 | 29 |
| Case 2: Ours | 0.59 | 0.58 | 0.13 | 30 |

## 5.3 ABLATION STUDIES

To further highlight the strength of the shared representations, we present the results of our model with ablations of the causal structure learning and outcome prediction components. We use the synthetic data described in Section 5.1 and report the MSE for regression on the outcome variable $Y$. As shown in Table 3, our model achieves the lowest MSE and demonstrates superior generalization performance in both cases.

## 5.4 SCALABILITY ANALYSIS

We compare the time complexity of our model with CASTLE. We use the synthetic dataset from Case 2 for the analysis. We measure the average training time across the 10 folds of both models and plot it against the number of variables $d$ (Figure 1). We also report the corresponding average test MSEs of the models in Table 4. Our model efficiently scales with the number of feature variables while consistently maintaining a stable MSE score. CASTLE uses one feed forward network for each feature variable and does not scale with the number of feature variables.

## 5.5 GENERALISATION PERFORMANCE ON REAL DATA

We also study the generalisation performance of the models on publicly available datasets from The UCI Machine Learning Repository (Markelle Kelly). We choose two binary classification datasets - Statlog Heart and Breast cancer (Wisconsin Diagnostic), and one multi-class classification dataset - Las Vegas ratings. We choose CASTLE as our primary baseline here because CASTLE also uses causal structure learning for outcome prediction and has outperformed state-of-the-art regularisation methods like dropout, data augmentation and batch normalisation for Statlog heart and Las Vegas ratings.

Table 3: Ablation results for causal structure learning and outcome prediction components using synthetic data. Test MSE for regression task along with the gap between train and test MSE ($\Delta :=$ Mean $\text{MSE}_{test}$ - Mean $\text{MSE}_{train}$) are reported.

|  | Case 1 | Case 2 |
|---|---|---|
| causal structure learning alone | $1.172 \pm 0.000 \ (0.164)$ | $1.027 \pm 0.000 \ (0.148)$ |
| outcome prediction alone | $0.951 \pm 0.041 \ (0.188)$ | $0.848 \pm 0.038 \ (0.088)$ |
| Ours | $\mathbf{0.938 \pm 0.029 \ (0.086)}$ | $\mathbf{0.815 \pm 0.029 \ (0.002)}$ |

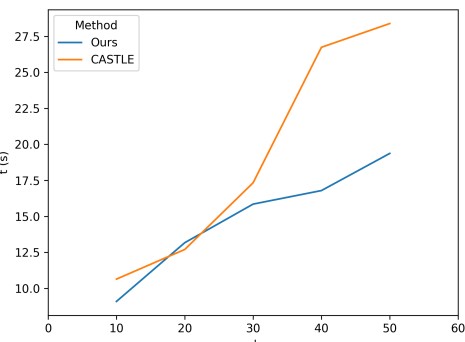

Figure 1: Comparison of average training time against the number of feature variables $d$.

Table 5 shows the performance of the models for the classification tasks. Area Under receiver operating characteristic Curve (AUC) is chosen as the evaluation metric. We see that on fairly simple binary classification datasets, all the models perform similarly and reach an AUC greater than 0.9. However, on the relatively difficult multi-class classification task, which involves the classification of the samples into 5 classes, MLP-based models and CASTLE fail to perform well on the test data. Our model performs consistently well on all the datasets. The ES variant performs early stopping based on validation accuracy in case of classification. Therefore, the performance of ES is worse than L2+ES in the multi-class classification task where accuracy might not be a robust metric. The recovered causal graphs from our model and dataset details are included in Appendix E.

### 5.6 CASE STUDY: APPLICATION TO SURVIVAL ANALYSIS

We choose the Worcester heart attack study dataset (Hosmer Jr et al., 2008) for our case study. The dataset was originally designed to study the trends in incidence rates and patient outcomes across multiple decades (Floyd et al., 2009). We use a publicly available subset of this dataset [1], which contains 500 samples collected across three years (1997, 1999, 2001). We choose death until the length of hospital stay as our endpoint. We convert the time-to-event analysis problem to a classification problem by predicting the likelihood of death before a specific length of hospital stay $= k$ (see Appendix F). We study two scenarios as follows.

**Scenario 1.** In this scenario, we randomly split the datasets into into 90% training and 10% test data stratified according to the labels. The training data is used in a 10-fold cross validation setting to train the models. The threshold $k$ for the endpoint is the median length of hospital stay.

---

[1] `https://web.archive.org/web/20170517071528/http://www.umass.edu/statdata/statdata/data/whas500.txt`

Table 4: Test MSE for varying number of feature variables $d$.

|        | CASTLE          | Ours            |
|--------|-----------------|-----------------|
| $d = 10$ | $0.667 \pm 0.027$ | $0.626 \pm 0.017$ |
| $d = 20$ | $0.923 \pm 0.089$ | $0.815 \pm 0.029$ |
| $d = 30$ | $1.104 \pm 0.100$ | $0.830 \pm 0.023$ |
| $d = 40$ | $1.305 \pm 0.159$ | $0.874 \pm 0.026$ |
| $d = 50$ | $1.018 \pm 0.087$ | $0.693 \pm 0.022$ |

Table 5: Classification: Generalisation performance on real-world datasets. Baseline: MLP; L2+ES: MLP + L2-norm with early stopping based on training loss; ES: MLP + L2 norm + early stopping based on validation accuracy; Test AUC along with the gap between train and test AUC ($\Delta$ := Mean $\text{AUC}_{train}$ - Mean $\text{AUC}_{test}$) are reported.

|          | Statlog Heart           | Breast cancer            | Las Vegas ratings          |
|----------|-------------------------|--------------------------|----------------------------|
| Baseline | $0.935 \pm 0.012$ (0.064) | $0.991 \pm 0.001$ (0.008) | $0.574 \pm 0.057$ (0.380)   |
| L2+ES    | $0.936 \pm 0.011$ (0.062) | $0.996 \pm 0.000$ (0.002) | $0.571 \pm 0.056$ (0.383)   |
| ES       | $0.954 \pm 0.010$ (0.067) | $0.995 \pm 0.007$ (-0.006) | $0.336 \pm 0.044$ (0.253)   |
| CASTLE   | $0.928 \pm 0.020$ (0.054) | $0.997 \pm 0.003$ (-0.003) | $0.553 \pm 0.067$ (-0.042)  |
| Ours     | $0.931 \pm 0.017$ (0.068) | $0.996 \pm 0.003$ (0.001) | **$0.658 \pm 0.049$ (0.239)** |

**Scenario 2.** Here we simulate the scenario in the real world setting by choosing the patients studied during the years 1997 and 1999 as our training data, and the patients studied during 2001 as our test data. The training data is used in a 10-fold cross validation setting to train the models. The threshold $k$ for the endpoint is the median length of hospital stay over the training data.

Table 6 presents the results of all the methods in both the scenarios. CASTLE, Baseline and ES fail to generalise to test data in both scenarios. L2+ES performs well when trained on data from scenario 1 but fails to generalise to test data in scenario 2. Our method performs the best in both scenarios confirming our hypotheses: (i) our method performs well in predicting outcome over evolving knowledge and treatment approaches; (ii) our method performs better than the baseline models in both scenarios emphasising the importance of proposed approach for generalisation;

Figure 2 illustrates the causal graph recovered from our models. The graph in Scenario 1 shows an association between death during hospital stay and the factors such as age, body mass index (bmi), atrial fibrillation (afb), cardiogenic shock (sho) and complete heart blockage (av3). In addition to age, bmi, sho and av3, the graph in Scenario 2 shows an association with initial heart rate (hr), order of myocardial infarction (miord) and congestive heart complications (chf). The robust performance of our model in both scenarios highlights the ability of our model to adjust to evolving real world scenarios.

# 6 CONCLUSION

We introduce a novel paradigm that leverages the best of both worlds - causal structure learning and machine learning-based outcome prediction for improved outcome prediction. Through experiments on several synthetic and real data, we demonstrated that this strategy not only improves out-of-sample generalisation for outcome prediction but also improves interpretability by learning causal structure. We also demonstrated advantages of our model with respect to robust causal discovery for

Table 6: Classification performance on Worcester heart attack study dataset. Baseline: MLP; L2+ES: MLP + L2-norm with early stopping based on training loss; ES: MLP + L2 norm + early stopping based on validation accuracy; Test AUC along with the gap between Train and test AUC ($\Delta :=$ Mean AUC$_{train}$ - Mean AUC$_{test}$) are reported.

|  | **Scenario 1** | **Scenario 2** |
|---|---|---|
| Baseline | $0.595 \pm 0.049$ (0.394) | $0.582 \pm 0.022$ (0.417) |
| L2+ES | $0.652 \pm 0.037$ (0.229) | $0.595 \pm 0.015$ (0.351) |
| ES | $0.499 \pm 0.020$ (-0.074) | $0.426 \pm 0.009$ (0.012) |
| CASTLE | $0.567 \pm 0.221$ (0.085) | $0.588 \pm 0.122$ (0.240) |
| Ours | $\mathbf{0.834 \pm 0.127\ (0.081)}$ | $\mathbf{0.693 \pm 0.120\ (0.237)}$ |

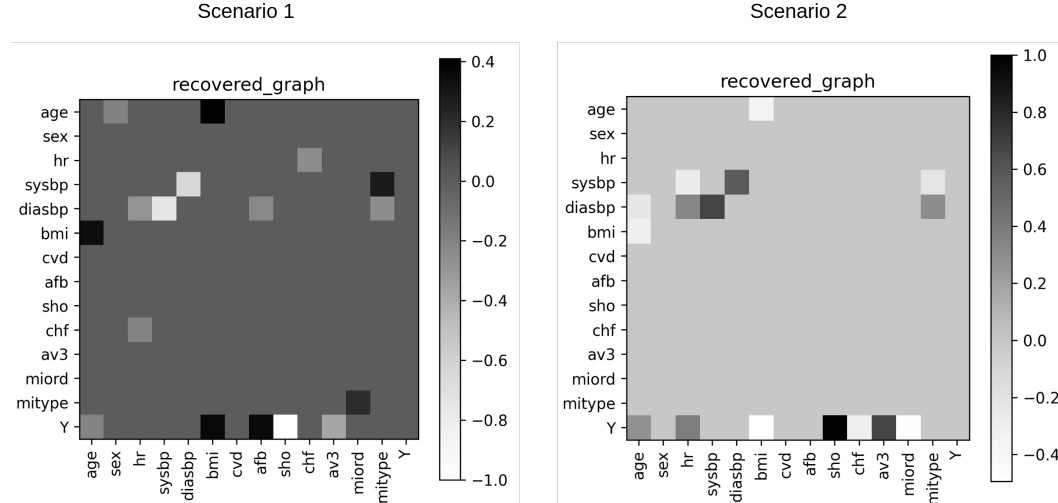

Figure 2: Causal graph recovered from our models in both scenarios of the experiments on Worcester heart attack study dataset.

the outcome variable and time efficiency. With a case study on survival analysis, we demonstrated the potential translational value that our model provides.

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

## A  AUGMENTED LAGRANGIAN METHOD-BASED OPTIMISATION

We use Augmented Lagrangian method to optimise the constrained optimisation problem in (5) with the acyclicity constraint $h(\mathbf{W}) = tr(e^{\mathbf{W} \odot \mathbf{W}}) - d$ as:

$$\mathcal{L}_\rho(\mathbf{W}, \Theta_1, \Theta_2, \Theta_3, \alpha) = \frac{(1-\kappa)}{2n} \sum_{i=1}^{n} \left\| X^{(i)} - \hat{X}^{(i)} \right\|_F^2 + \lambda \left\| \mathbf{W} \right\|_1 + \alpha h(\mathbf{W}) + \frac{\rho}{2} |h(\mathbf{W})|^2$$

$$+ \frac{\kappa}{n} \sum_{i=1}^{n} \left\| Y^{(i)} - \hat{Y}^{(i)} \right\|_F^2,$$

where $\alpha$ is the Lagrange multiplier, $\rho > 0$ is multiplier for penalty and $\lambda$ is the L1-norm penalty for $\mathbf{W}$. We solve the following optimisation problem by using Adam optimiser at each iteration:

$$\mathbf{W}^{k+1}, \Theta_1^{k+1}, \Theta_2^{k+1}, \Theta_3^{k+1} = \underset{\mathbf{W},\Theta_1,\Theta_2,\Theta_3}{\arg\min} \mathcal{L}_\rho^k(\mathbf{W}, \Theta_1, \Theta_2, \Theta_3, \alpha^k)$$

We then update the parameters $\alpha$ and $\rho$ for the next iteration as:

$$\alpha^{k+1} = \alpha^k + \rho^k h(\mathbf{W}^{k+1}),$$

$$\rho^{k+1} = \begin{cases} \beta\rho^k, & \text{if } |h(\mathbf{W}^{k+1})| \geq \gamma|h(\mathbf{W}^k)|, \\ \rho^k, & \text{otherwise}, \end{cases}$$

where $\gamma < 1$ and $\beta > 1$ are training hyperparameters.

## B    REPRODUCIBILITY STATEMENT

All the methods used Adam optimiser with a learning rate of 1e-3. The models were trained for 300 epochs with an early stopping criterion based on validation loss (except the L2-norm with early stopping (ES) variant of the MLP which employed early stopping based on validation score). In addition, our model and CausalGAE update Lagrange multiplier $\alpha$ and penalty $\rho$ over 20 iterations with early stopping based on a threshold. For both models, we use the default threshold and loss hyperparameters as in (Ng et al., 2019). All methods used the same data splits and identical seeds. For all our experiments we use a machine equipped with Intel i9-10900X processor and NVIDIA RTX2080 GPUs. We intend to make our code publicly available.

## C    SENSITIVITY ANALYSIS

We perform a sensitivity analysis for the loss hyperparameter $\kappa$ by using the synthetic datasets in Case 1:

$$\mathbf{X} = 2\sin(\mathbf{W}^T(\mathbf{X} + \mathbf{1}) + 0.5 \cdot \mathbf{1}) + \cos(\mathbf{W}^T(\mathbf{X} + \mathbf{1}) + 0.5 \cdot \mathbf{1}) + \mathbf{Z}$$

and Case 2:

$$Y = 2\sin(\mathbf{W}^T(\mathbf{X} + \mathbf{1}) + 0.5 \cdot \mathbf{1}) + \cos(\mathbf{W}^T(\mathbf{X} + \mathbf{1}) + 0.5 \cdot \mathbf{1}) + \mathbf{Z}$$
$$+ \cos((X^{non-pa(Y)_0} + \mathbf{1}) + 0.5 \cdot \mathbf{1})$$

This hyperparameter controls the fraction of supervised loss added to the overall loss function. Figure 3 illustrates the test MSE loss with respect to $\kappa$. We observe minimal variation between $\kappa = 0.25$ and $\kappa = 0.75$. The MSE loss is worst at $\kappa = 0$, reinforcing the importance of supervised loss for generalisation. We choose $\kappa = 0.25$ for all our experiments.

## D    ROBUST CAUSAL DISCOVERY

Figures 4 and 5 compare the true causal graph with causal graphs recovered from CausalGAE and our model for synthetic datasets from Case1 and Case 2 respectively. In contrast to our model, CausalGAE fails to identify the causal parents of $Y$ in both the cases.

## E    CLASSIFICATION ON REAL DATA

We show the causal graphs recovered by our model for various real datasets here (Figure ). The Statlog heart dataset has 270 samples and 13 features. The Breast cancer (Wisconsin Diagnostic) dataset includes 569 samples and 30 features. The Las Vegas ratings dataset includes 504 samples and 19 features.

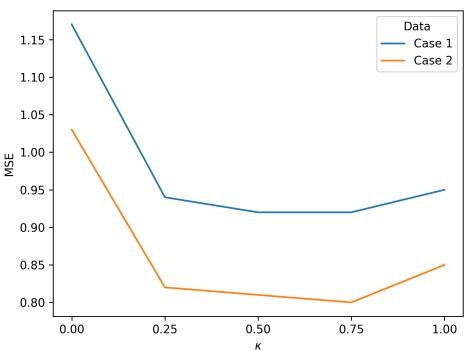

Figure 3: Sensitivity of our model performance to the loss hyperparameter $\kappa$.

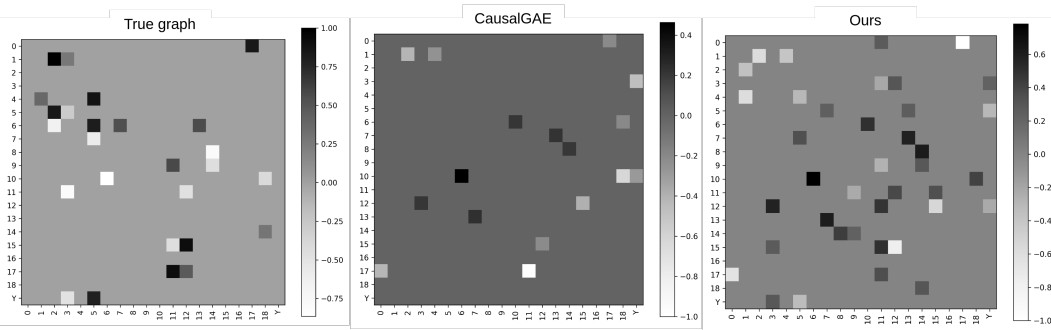

Figure 4: True causal graph for synthetic data Case 1 and the recovered graphs from CausalGAE and our model.

# F APPLICATION TO SURVIVAL ANALYSIS

We convert the time-to-event analysis problem to classification by thresholding the continuous time and assigning ground truth labels based on the event. Specifically, a positive class label is assigned to those cases where an event (death at discharge) occurred before $k$ days, $k$ being the median length of hospital stay in Scenario 1 and median length of hospital stay over the training cohort in Scenario 2. A negative label is assigned for all other cases where no event occurred before $k$ days.

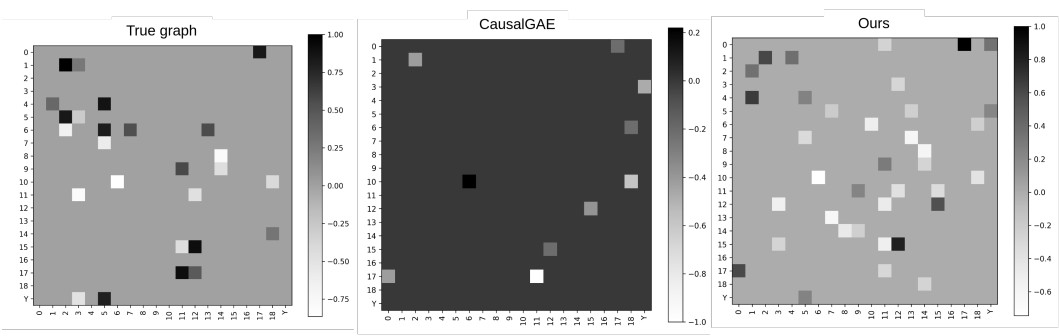

Figure 5: True causal graph for synthetic data Case 2 and the recovered graphs from CausalGAE and our model.

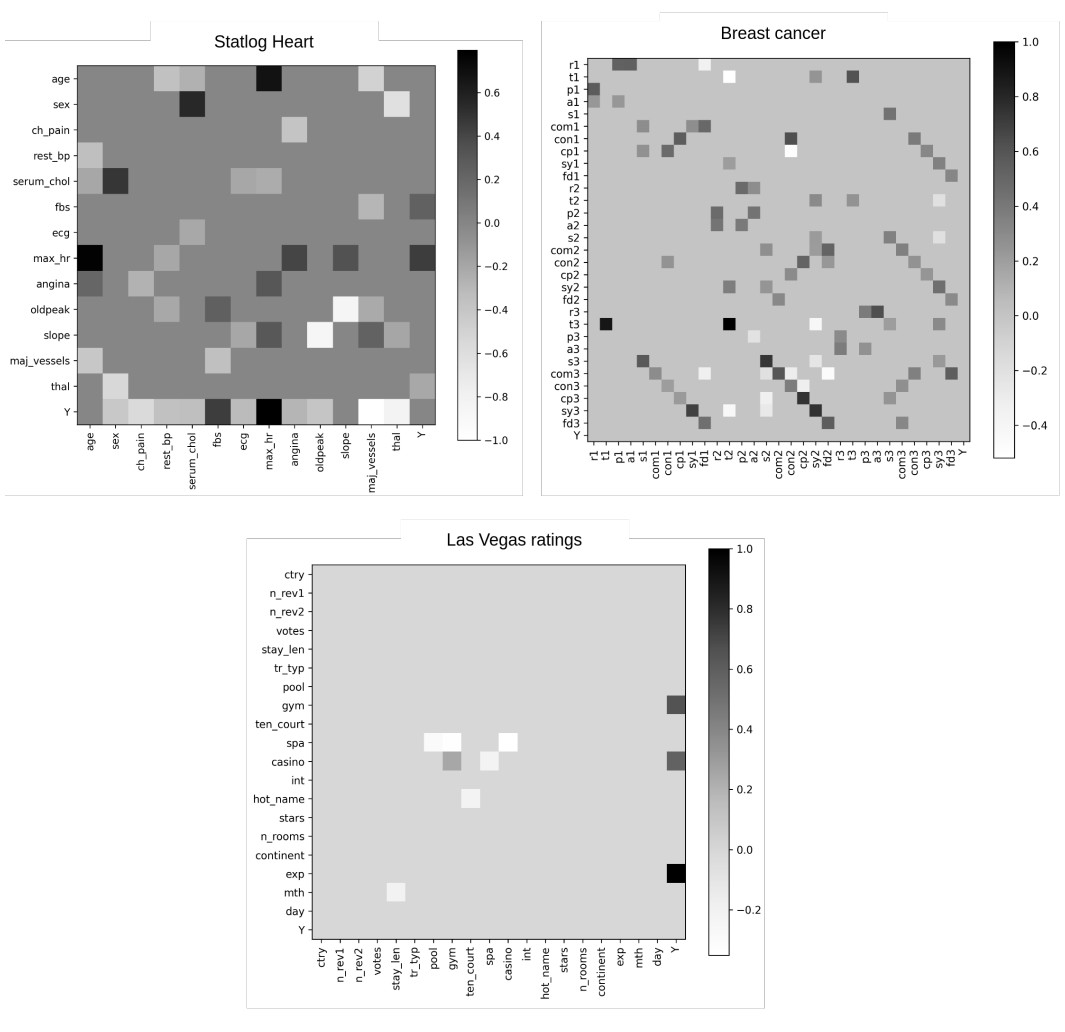

Figure 6: Causal graphs recovered from our model for classification on real datasets.

