# OpenReview forum: "The best of both worlds: Improved outcome prediction using causal structure learning"
_ICLR.cc/2025/Conference — ICLR 2025 Conference Withdrawn Submission_

### Official Review · Reviewer_iYup · 2024-10-22

**Soundness:** 2
**Presentation:** 2
**Contribution:** 2
**Rating:** 3
**Confidence:** 4

**Summary:**

The paper introduces an approach that combines causal structure learning with machine learning-based outcome prediction, aiming to improve out-of-sample generalization and interpretability. While this concept is potentially useful, the execution and justification in the paper are insufficient. The paper fails to address several critical issues in methodology, empirical evaluation, and clarity of presentation, which significantly weakens its contribution and impact on the field. As a result, the paper does not meet the necessary standard for acceptance.

**Strengths:**

1. Conceptual Motivation: The idea of integrating causal structure learning with outcome prediction to improve generalization is theoretically appealing, especially in domains like healthcare where data are limited.

2. Scalability Focus: The authors attempt to address the scalability of the model, which is an important concern in high-dimensional datasets, but the results remain underexplored.

**Weaknesses:**

1. Weak Empirical Results: The performance improvements claimed by the paper are not convincingly demonstrated. The results, especially on real-world datasets, are marginal and do not justify the complexity of the approach. Furthermore, the paper lacks proper evaluation against stronger baselines or state-of-the-art methods in the field.

2. Lack of Reproducibility: The implementation details are inadequate for reproducing the study, with little attention given to hyperparameters, model training configurations, or specific choices that lead to the presented results. The absence of detailed code or guidelines for reproduction is a serious concern.

3. Unclear Methodology: The mathematical derivations and technical explanations are convoluted and difficult to follow, making it hard to understand the key contributions of the model. Important aspects, such as how causal discovery improves prediction, are not well explained.

4. Lack of Real-World Relevance: Although the method is intended for use in domains such as healthcare, the paper does not adequately discuss the practical challenges of applying the model in real-world settings. Factors such as computational efficiency, data quality, and adaptability to evolving medical knowledge are not sufficiently addressed.

**Questions:**

Given the paper’s significant shortcomings in empirical validation, clarity, and reproducibility, it is not suitable for acceptance in its current form. Below are detailed suggestions on how the authors could address the identified weaknesses:

For rebuttal, the authors should:
1. Improve Reproducibility (Experimental Setup): The authors should provide more comprehensive implementation details directly in the paper, not just in appendices or supplementary materials. For example, the hyperparameter tuning details, such as how the parameter κ in Equation (5) is selected and tuned, should be included in the Results section.

2. Enhance Comparisons to Baselines (Section 5): The authors should consider more advanced causal learning models and baseline prediction methods in Section 5.1. They should justify why CASTLE, MLP, and L2+ES are chosen as primary baselines and compare their method to more recent models. For instance, they should explore recent deep learning-based models, such as graph-based neural networks (GNNs), for causal discovery and models based on temporal attention for healthcare predictions.

3. Address Practical Challenges in Real-World Application (Section 5.5): The authors should address how their method would perform with incomplete, imbalanced, or noisy data—common issues in real-world healthcare. Furthermore, a detailed discussion of the computational cost and the resources required to run the proposed model in large-scale clinical settings should be added to the Scalability Analysis section. This would make the paper more practically relevant to real-world medical applications.

4. Conduct Ablation Studies (Section 5): To better understand the contribution of each component of the proposed method, the authors should perform ablation studies. For example, in Table 5, how does the model perform when the causal structure learning component is removed? These studies would clarify whether causal structure learning actually contributes significantly to outcome prediction.

5. Deepen Analysis of Results (Section 5): The Results section, especially Table 5 (Worcester heart attack study), should include more detailed discussions of why the method performs better than baselines. What specific features or causal relationships drive this improvement? A more nuanced interpretation of the results would strengthen the paper’s empirical contributions.

---

### Official Review · Reviewer_Ttk2 · 2024-10-28

**Soundness:** 2
**Presentation:** 2
**Contribution:** 1
**Rating:** 5
**Confidence:** 2

**Summary:**

This paper proposes a hybrid approach that combines causal structure learning with outcome prediction, aiming to enhance out-of-sample generalization in predictive tasks and improve interpretability through learning causal structures.

**Strengths:**

1. Clever idea of adding an outcome term in the objective function, combining causal structure learning with outcome prediction.
2. The real-world survival analysis case study highlight the practical potential of the method in medical context with limited data.

**Weaknesses:**

1. The paper mainly focuses on outcome prediction accuracy; however, since it also claims to learn causal structure, it would be valuable to see a comparison with other baseline structure learning algorithms. The causal graph in Figure 2 (line 411) lacks clarity and does not effectively illustrate structure learning performance—it appears more suggestive of correlation rather than causation.
2. In line 160, what is the dimension of Y? If i understand correctly, Y is a single label but can have multiple classes. Could the outcome Y also represent several different labels, each with multiple classes? Additionally, can the model handle this type of multi-label, multi-class task?
3. While the model is designed for medical data, can it generalize to other fields? For example, could it handle extremely high-dimensional data with sparse and noisy features, such as genomic datasets?

**Questions:**

See Weaknesses.

---

### Official Review · Reviewer_b8yN · 2024-10-30

**Soundness:** 1
**Presentation:** 3
**Contribution:** 2
**Rating:** 1
**Confidence:** 4

**Summary:**

The authors present a method that purports to learn causal structure while optimizing outcome prediction.

**Strengths:**

A strength of this paper is that several empirical examples are taken up and analyzed, showing some incremental advantages.

**Weaknesses:**

This is the start of a good paper, but several issues need to be addressed. I'll list a few.

The main weakness of the paper is the absence of a theoretical argument for correctness. The method is presented very quickly, but no assurance is given that convergence will happen, or that convergence to a correct causal graph will happen, even in the large sample limit. The argument, rather, relies on an extended series of examples showing incremental improvement. The examples are interesting, but more is required. There is considerable theory at this point for inferring causal graphs using soft acyclicity constraints; this is not referred to, taken up, or responded to.

In addition, a whole host of methods for causal search are available in the literature, and various types of theory are used. This literature needs to be reviewed, and reasons should be given as to why other such methods are not compared. Standard constraint-based methods could be used for this purpose, for instance, and if outcome predictions need to be optimized, parameters can be adjusted for that purpose. What is the _advantage_ of doing it in the proposed way? This is unclear. It is important that novel techniques are shown to be clear steps forward in the literature.

Only a very restricted set of nonlinear functions is taken up for the nonlinear simulation examples. This could be expanded, or a more general method of simulating nonlinear functions could be employed.

Finally, as mentioned the results are incremental improvements over previous results. One gets the impression that with some additional theoretical and coding work, results could be obtained that improve over previous results in significant ways. Anyway, that is an aspiration.

**Questions:**

The paper needs arguments and theorems to show theoretical soundness. Could you take this up?

One may get the impression that the authors are not familiar with causal theory more generally, outside of the soft acyclicity optimization ideas, and perhaps not even for that entire literature. Could you please add text to assure the readers that you have carefully considered alternative ways of proceeding and are offering your way as an improvement over previous techniques for this purpose?

The simulation section can be expanded to include a wider variety of nonlinear functions. One may get the impression that the simulations work just for these nonlinear functions and not for others. Could this be expanded?

---

### Official Review · Reviewer_eWJU · 2024-11-02

**Soundness:** 1
**Presentation:** 2
**Contribution:** 1
**Rating:** 3
**Confidence:** 4

**Summary:**

Authors propose a novel causal discovery algorithm via continuous optimization capable of learning a DAG and predicting the outcome variable jointly.

**Strengths:**

- The proposal of jointly learning and predicting the relevant outcome.

**Weaknesses:**

- The contributions stated at the beginning of the paper are confusing: I'm unsure what "shared representations of the data" even mean.
- The improvement w.r.t. CausalGAE is marginal.
- The SHD in Table 2 is worse than existing techniques: the predicted DAG is far from the true DAG. The metrics in Table 3 and Table 4 show little or no improvement over existing techniques.
- The proposed approach is compared against only two baselines, with very similar results.
- The study case about survival has no causal analysis at all.
- There is no causal interpretation of the learnt DAGs.

**Questions:**

- What does "shared representations of the data" mean?
- How the "visualisation of the learnt causal graph" is an improvement over the existing causal discovery techniques?
- Why is "outcome prediction" performed using causal graphs? It is known that causal graphs are not useful when it comes to prediction tasks: they are useful for causal inference.
- Where is the rho parameter in Equation 6?
- Why is there a task about classification? How should this prove that your technique generalize? This only shows that the learnt graphs are somewhat able to generalize, it tells nothing about your technique ability to generalize.
- Why the survival analysis case is discussed in light of association rather than causation if this paper is about causality?

---

> ### Author Response · Authors · 2024-11-19
>
> Thank you for the review and valuable feedback. In the latest revision, we have clarified the main focus of our work. Below, we address the open questions and provide responses to the reviewer’s comments. Comments in the weaknesses section are labelled as W1, W2, etc., corresponding to the respective reviewer remarks. Similarly, answers to the questions are labelled Q1, Q2, Q3, etc., corresponding to the reviewer’s queries. We hope that these clarifications address the reviewer’s concerns.
>
> W1) The primary focus of our work is to improve generalisation for outcome prediction in the medical domain. As causal structure learning has the potential to discover reliable associations among the feature variables and enhance interpretability, we designed our approach to learn outcome prediction and causal structure simultaneously. In our framework, causal structure learning functions as an auxiliary task to support outcome prediction, sharing representations of the input through the hidden layers of our network architecture and employing task-specific heads for refined predictions.
>
> W2 & W3) The primary focus of our work is to improve generalisation for outcome prediction in the medical domain. Causal structure learning functions as an auxiliary task to support outcome prediction. Despite this, the results in Table 2 show an improvement in the True Positive Rate (+35% for both cases of synthetic data). Also, CausalGAE [1] fails to identify the causal parents of Y in both the cases as seen in the associated causal graphs included in Appendix D.
>
> W3) For outcome prediction, we choose CASTLE [2] as one of our baseline models, because CASTLE [2] also uses causal structure learning for outcome prediction.  CASTLE [2] has outperformed state-of-the-art regularisation methods like dropout, data augmentation and batch normalisation for two of the datasets we use in our study - Statlog heart and Las Vegas ratings. Besides CASTLE, we also compare our method with regularisation techniques like early stopping and L2 penalty which are still common approaches to improve generalisation for outcome prediction.
>
> Table 4 includes results from two binary classification datasets and one multi-class classification dataset. For the binary classification tasks, the results of all models are near saturation, achieving performance close to 100%, indicating that these tasks are relatively simple. In contrast, the multi-class classification task, which requires classifying samples into five categories, is more challenging. On this task, our model demonstrates a significant improvement, achieving a +10.5% higher AUC compared to CASTLE.
>
> Table 3 supports our scalability analysis. Along with Figure 1, Table 3 illustrates that our model efficiently scales with the number of feature variables while consistently maintaining a stable mean squared error (MSE). Across different feature dimensions ($d={10, 20, 30, 40, 50}$), our model consistently outperforms CASTLE, with MSE reductions ranging from 0.04 to 0.43.
>
> Based on these results, we respectfully disagree with the reviewer's comment, as the evidence demonstrates that our model achieves superior performance both in terms of classification accuracy and scalability.
>
> W5 & W6) In the case study on survival analysis, the ground truth causal graph is not available. However, we visualise the learned directed acyclic graph (DAG) and provide our commentary on it in lines 416–421. We appreciate the feedback and will elaborate further on this aspect in the next revision of our paper.
>
>
> Answers to the questions:
>
> Q1) We do not claim that the visualisation of the learned causal graph represents an advancement over existing causal discovery techniques. Instead, we assert that the visualisation of the learned DAG enhances the interpretability of the outcome prediction task.
>
> Q2) The primary focus of our work is to improve generalisation for outcome prediction in the medical domain. Causal structure learning serves as an auxiliary task to support outcome prediction. This is achieved by sharing representations of the input through the hidden layers of our network architecture and employing task-specific heads for refined predictions. Additionally, we did not find strong evidence to suggest that causal graphs are not useful for prediction tasks. We would greatly appreciate it if the reviewer could share relevant literature supporting this perspective. Thank you.
>
> Q3) Thank you for your valuable feedback. Equation 6 is a condensed representation of a series of equations provided in Appendix A, where the parameter $\rho$ is explicitly included. To avoid potential confusion, we have removed the explanation of this parameter from line 201.
>
> (Part 1/2)

---

> > ### Author Response · Authors · 2024-11-19
> >
> > Q4) In our framework, causal structure learning functions as an auxiliary task to support outcome prediction, sharing representations of the input through the hidden layers of our network architecture and employing task-specific heads for refined predictions. Outcome prediction can be considered a form of classification, especially in the context of categorical outcomes. If this explanation does not fully address the reviewer’s concern, we would appreciate further clarification on the specific point they are asking about.
> >
> > Q5) In the case study on survival analysis, the ground truth causal graph is not available. Therefore, we use the term 'association' rather than 'causation' to accurately reflect the nature of the relationships in the data. This distinction ensures that our results remain consistent with the data and the available information.
> >
> > References:
> > [1] Ignavier Ng, Shengyu Zhu, Zhitang Chen, and Zhuangyan Fang. A graph autoencoder approach to causal structure learning. arXiv preprint arXiv:1911.07420, 2019.
> > [2] Trent Kyono, Yao Zhang, and Mihaela van der Schaar. Castle: Regularization via auxiliary causal graph discovery. Advances in Neural Information Processing Systems, 33:1501–1512, 2020.
> >
> > (Part 2/2)

---

### Official Review · Reviewer_6ML7 · 2024-11-03

**Soundness:** 1
**Presentation:** 2
**Contribution:** 1
**Rating:** 3
**Confidence:** 4

**Summary:**

The work attempts to use causal structure learning method to improve out-of-sample generalisation. The work is weill-organized and easy to understand.

**Strengths:**

In this paper, the authors propose to leverage causal structure leanring into prediciton problem for improve improve out-of-sample generalisation. The idea is interesting.

**Weaknesses:**

The idea of leveraging causal structure leanring into the machine generalisation problem is not new and lots of work have been proposed for addressing this problem. The weakness and questions are as follows.

In the abstract, the authors mentioned that “due to evolving conditions and treatment approaches, causal relationships between the variables change over time”, but in this work, the authors do not give any solutions to solve this problem any more.

By reading the abstract and the introduction, it is clear which problems authors attempt to address, causal effect problem or machine learning generalization problem? For one of the problems, authors do not clearly give the key Pros and Cros of existing methods for a strong motivation. Specifically, if the work is for the causal effect problem, the authors should give the detailed discuss the state-of-the-art causal effect estimation methods, while for machine learning generalization problem, the authors should discuss deeply existing model generalization algorithms.

In addition, the confounder problem is a long-standing problem in causal inference, from the introduction, it is hard to find any novel idea proposed in the work to address this problem.

Eq.(2) is only a general latent representation of input data X. How can Eq.(2) restrict  predicting Y using a non-linear function of its causal parents?

Eq.(3) only simply replaces the WX in Eq.(1) with a latent representation by neural networks without providing any novel contributions.

In Eq.(5), the first two parts aims to learn a DAG, it is hard to make Eq.(5) focus on the causal variables or important ones (such as parents) for predictions.

In Eq.(5), how do the authors address the confounder problem?

In experiments, (1) the used datasets are too small with only no more than 100 features; (2) many benchmark OOD datasets are not used; (3) many state-of-the-art machine learning generalization are not compared and discussed in the related work and in the experiment part.

**Questions:**

In the abstract, the authors mentioned that “due to evolving conditions and treatment approaches, causal relationships between the variables change over time”, but in this work, why do the authors not give any solutions to solve this problem any more?

Eq.(2) is only a general latent representation of input data X. How can Eq.(2) restrict  predicting Y using a non-linear function of its causal parents?

Eq.(3) only simply replaces the WX in Eq.(1) with a latent representation by neural networks, lots of work has used this idea, what are the novel contributions in the work?

In Eq.(5), the first two parts aims to learn a DAG,  how can Eq.(5) focus on the causal variables or important ones for predictions?

In Eq.(5), how do the authors address the confounder problem?

In experiments, why do the authors not compare their algorithm with the latest  out-of-sample generalisation with the widely used benchmard OOD datasets?

---

### Official Review · Reviewer_bp78 · 2024-11-04

**Soundness:** 1
**Presentation:** 2
**Contribution:** 2
**Rating:** 3
**Confidence:** 4

**Summary:**

This paper proposes to use a model for both the task of outcome prediction and causal discovery, by using a shared weight weighted matrix from causal graph and two headers for the two tasks correspondingly. This paper claims that using this framework can result in enhanced performance in both tasks. In particular, it can make causal discovery more robust to unmeasured confounders. The performance is demonstrated through simulation and real data.

**Strengths:**

- The idea of bringing in outcome prediction into causal discovery framework is novel
- Multiple datasets are included in real study session

**Weaknesses:**

- No intuition or theorem is shown for why including the outcome prediction components in causal discovery helps improve the performance in causal graph learning.

- Simulation settings are limited: for functional causal model, only an ANM with sin() and cos() function and its variant with non-parent effect (case 2) is considered. For other parameters such as noise distribution, graph type, and edge density, there seems to be only one setting

- Not many basline causal discovery methods are compared with the proposed method. When compared with CASTLE, there is no improvement on FDR, and the performance seems to be worse according to SHD


- For time cost of the proposed method, no intuition is provided on why the proposed method is more efficient. No time complexity order is given

**Questions:**

- Following Weakness point 1, why would including the prediction loss in equition (5) helps causal discovery? Can authors provide some intuitive explanation or theorems to explain the results?

- Will including outcome prediction components in model always helps causal discovery? Will including the outcome prediction component sometimes cause model to be biased to the prediction task, leading to worse causal graph learning result? If this is true, can authors show an ablation study of when outcome prediction component helps and when outcome prediction component doesn't help?

- It seems case 2 of simulation doesn't really represet a case with unmeasured confounder. In case 2 , authors claim that "the outcome is not dependent only on its causal parents" due to that $X^{non-pa(Y)}$ term. However, this is because authors still treat the weighted matrix W as the true causal graph. Actually, when authors generate the Y variable based on  $X^{non-pa(Y)}$, the original $W$ is no longer represents the true underlying graph. Rather the weighted adjacency of the true causal graph should be updated to $W^{new}$ such that its entry corresponding to $X^{non-pa(Y)}, Y$ needs to be filled with $1$ rather than $0$. Still, $Y$ is generated by the observed variables in $X$, not some unobserved variables not provided to causal discovery. Therefore, I don't think this is a case of "unmeasured confounder", nor can it prove the proposed method's robustness to assumption violation

- In simulation setting case 1 and case 2, what is the distribution of noise vector Z? The settings of generating graphs and data are rather limited. Can authors includem more settings of functional type (such as linear model or other non-linear model in ANM),  noise distribution, graph type, and edge density to demonstrate the proposed method can succeed not only in the limited cases presented in Section 5?


- In results of Section 5.2, can authors provide the standard error of the causal discovery evaluation metrics? Also, the proposed method seem to perform worse according to FDR and SHD, which contradicts the claim that the proposed method improves causal discovery, can authors explain that?

- In Section 5.2, can authors compare with more causal discovery methods (for example, some classific methods from the 3 classes reviewed in Section 2)?

- In Section 5.3, can authors provide intuition of why the proposed method is more efficient? In fact, the proposed method optimizes on both causal discovery loss and the prediction loss, but ends up with faster convergences, this could be counter intuitive.  Is it possible to prove time complexity of proposed method follows smaller order compared to CASTLE?

- Minor points on paper writing: In Section 3, why would authors mention "linear structural equation model" while the simulation setting doesn't follow LSEM?

---

### Author Response · Authors · 2024-11-18

Thank you for the reviews and valuable feedback.  We are grateful to the reviewers for appreciating the idea behind our work. The primary focus of our study was to improve out-of-sample generalisation for outcome prediction in the medical domain, and we have clarified this aspect in the latest revision. We hope these revisions address the reviewers' comments regarding the performance of causal discovery, and demonstrate its role in supporting outcome prediction.

---

### Note · Authors · 2024-11-28

**Comment:**

We are grateful for the feedback and insights provided by the reviewers, which have helped us reflect on ways to strengthen our work. While our revisions clarified the main focus of the study and extended the analyses to emphasise the strengths of our method, we have identified an opportunity to enhance the impact of our findings through additional clinical analyses.
To fully realise the potential of our approach and present it in the strongest possible light, we have decided to withdraw our submission. This will allow us to incorporate these further analyses and deliver a more comprehensive and impactful manuscript in the future.
Thank you for your time and consideration.

**Withdrawal Confirmation:**

I have read and agree with the venue's withdrawal policy on behalf of myself and my co-authors.